# Enhancement of visual biological motion recognition in early-deaf adults: Functional and behavioral correlates

**Marie Simon**[1]*, **Latifa Lazzouni**[1], **Emma Campbell**[1], **Audrey Delcenserie**[1,2],
**Alexandria Muise-Hennessey**[3], **Aaron J. Newman**[3], **François Champoux**[2,4],
**Franco Lepore**[1]

**1** Département de Psychologie, Centre de recherche en neuropsychologie et cognition, Université de
Montréal, Québec, Canada, **2** École d'orthophonie et d'audiologie, Université de Montréal, Montréal,
Québec, Canada, **3** Department of Psychology and Neuroscience, NeuroCognitive Imaging Lab, Dalhousie
University, Halifax, Nova Scotia, Canada, **4** Centre de recherche de l'Institut Universitaire de Gériatrie de
Montréal, Montréal, Québec, Canada

* marie.simon@umontreal.ca

CHILE

**Data Availability Statement:** Behavioral Dataset
has been shared with a public repository: Simon,
Marie (2020): DATASET.xlsx. figshare. https://
figshare.com/articles/_/12081405.

## Abstract

Deafness leads to brain modifications that are generally associated with a cross-modal
activity of the auditory cortex, particularly for visual stimulations. In the present study, we
explore the cortical processing of biological motion that conveyed either non-communicative
(pantomimes) or communicative (emblems) information, in early-deaf and hearing individu-
als, using fMRI analyses. Behaviorally, deaf individuals showed an advantage in detecting
communicative gestures relative to hearing individuals. Deaf individuals also showed signifi-
cantly greater activation in the superior temporal cortex (including the planum temporale
and primary auditory cortex) than hearing individuals. The activation levels in this region
were correlated with deaf individuals' response times. This study provides neural and
behavioral evidence that cross-modal plasticity leads to functional advantages in the pro-
cessing of biological motion following lifelong auditory deprivation.

## Introduction

An increasing number of studies suggest that early sensory loss leads to the enhancement of
the other intact sensory modalities [1]. Several behavioral studies have shown that early-deaf
people possess enhanced abilities for visual localization and visual motion detection [2].
According to functional neuroimaging studies, the visual enhancements in early-deaf individu-
als are generally attributed to the recruitment of the deafferented auditory cortex [3–6].
Therefore, the visual crossmodal activity of the auditory cortex is typically defined as compen-
satory, meaning that deaf people rely more on their intact visual system to encode their envi-
ronment in comparison to hearing individuals [7]. Some tactile [8–11] and language abilities
(i.e., sign language and/or lip-reading) [12–17] are also associated with the recruitment of the
auditory cortex in deaf people [1] and support the compensatory reorganization of the brain
after early auditory deprivation.

**Funding:** This research was supported by the Canada Research Chair Program (#RGPIN-8245-2014, F.L.), the Canadian Institutes of Health Research (#166197, F.L.), and a grant from Med-EL Elektromed (A.J.N., F.L., and F.C.) and from Quebec Bio-Imaging Network (M.S)." The funders had no role in study design, data collection and analysis, decision to publish, or preparation of the manuscript.

**Competing interests:** The authors have declared that no competing interests exist.

This study's aim is to tackle the relevant topic of visual crossmodal plasticity following early auditory deprivation with the visual ability to perceive biological motion i.e. gesture sequences that characterize all living things [18]. The study of biological motion is an interesting issue since with only minimal pieces of visual information, such as point-lights at the main joints of the human body, people can efficiently recognize human actions [18,19]. Human movement recognition is essential for social cognition and interaction. With this ability, people can understand the gestural intentions of others and respond adequately [20]. For the deaf individuals using sign language, the adequate comprehension of human action is specifically critical to rapidly detect the presence of linguistic movements [21]. More generally, the ability to quickly recognize human motion also represents additional visual cues for deaf individuals to interpret their environment despite the auditory deprivation [22].

Originally, the cerebral network associated to the understanding of action (biological movement, human action) was referred to as the mirror neurons system [23]. The human mirror neurons system is formed by the inferior parietal lobule (IPL), the ventral premotor cortex (PMv) as well as the inferior frontal gyrus (IFG, BA 44/45) in the homologous brain of the macaque [24]. Henceforth, it is commonly accepted that the cerebral network of action understanding in humans is broader than the previously cited regions and also includes the posterior superior temporal sulcus (pSTS), the supplementary motor area (SMA, BA 6), the primary somatosensory cortex (S1, BA 1/2), the intraparietal cortex (IPS), the posterior middle temporal gyrus (pMTG) at the transition to visual area V5, and fusiform face area/fusiform body area (FFA/FBA) [25]. It is interesting that the neural responses associated to point-light biological motion recognition involve the same characteristic set of regions implicated in human action recognition [26].

In prior studies, several stimuli have been used to disentangle cerebral networks involved in either or both sign language and human action recognition processes between deaf native signers and hearing individuals. Among the human actions, meaningless gestures, pantomimes, emblems, and signs are conceptualized as a continuum in terms of linguistic properties, conventionalization, and semiotics characteristics [27]. Pantomimes are non-communicative gestures that are oriented towards an object, an action or an event [28] who can convey meaning on their own without speech [27]. Emblems are conventional communicative gestures [27] that are culturally influenced [29] and defined as non-verbal action used to convey information to others [30] (for illustration see Fig 1). These two types of gesture are not language per se. They differ from sign languages since the latter are natural human languages that have evolved spontaneously in Deaf communities, and possess all of the linguistic structural properties and complexity of spoken languages [31]. Although sign languages use the visual-manual rather than aural-oral modalities, the networks of brain regions recruited for spoken and signed language processing are largely overlapping [32].

To date, all of the previous studies on deaf signers fail to converge neuroimaging with behavioral results. Using fMRI, two studies have investigated the cerebral network involved in the passive observation of pantomimes by deaf native signers. These studies report a hypoactivation of the human mirror neuron system in the IFG, and the IPL in deaf signers individuals [21,33]. On the other hand, some neuroimaging studies with pantomimes [34], sequences of meaningless gestures [35,36] or a single emblem [37,38] support a similar human action network between the deaf signers and hearing individuals. In the current study, we attempt to replicate and extend these findings to multiple emblems. This way, the present study offers a robust comparison of the human action network between emblems and pantomimes. Indeed, these two stimuli differ according to whether they aim to transmit information or not, since emblems represent communicative gestures whereas pantomimes represent non-communicative gestures [19]. Additionally, only the emblems show some linguistic properties, such as

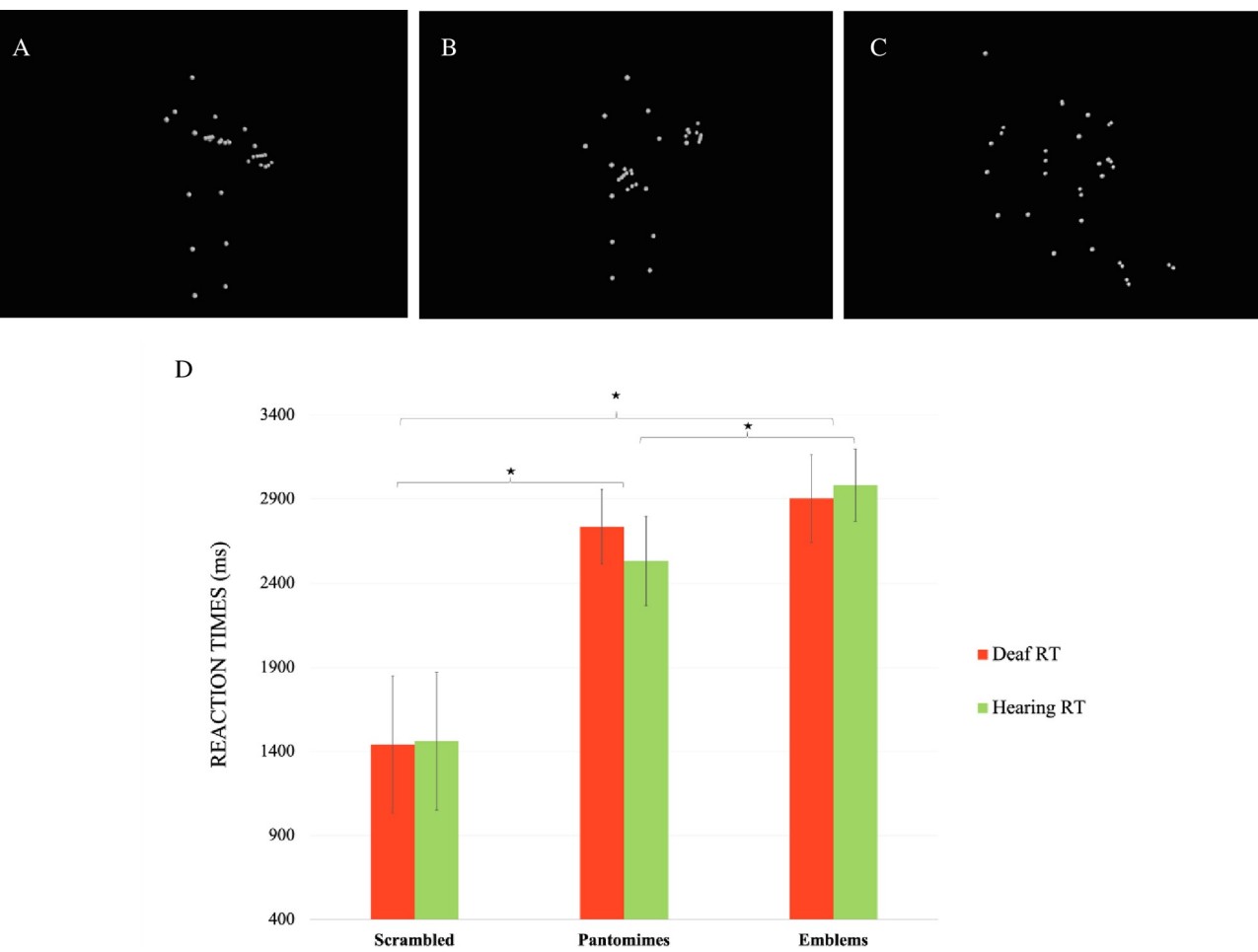

**Fig 1. Stimuli and behavioural results.** (A) example of a communicative gesture/emblems « calm down » (B) example of a non-communicative gesture/pantomimes « playing guitar » (C) example of a scrambled versions (D) Behavioral results illustrating the reaction times (RT) according to both groups. Errors bars denote standard deviation.

phonological and morphological components [27]. Furthermore, activation of the superior temporal gyrus (STG), including the primary auditory cortex and the planum temporale, has been observed across tasks requiring to recognize emblems, pantomimes, and meaningless gestures [14,34,37] despite the absence of behavioral differences in terms of accuracy or reaction time between deaf signers and hearing individuals. Together, these findings suggest that lifelong deafness and/or sign language use could lead to alterations in the neural networks recruited to interpret manual communication, even when it is not linguistically structured. Furthermore, increased recruitment of traditionally auditory and language processing areas during gesture recognition may reflect that lifelong reliance on visual communication (sign language and lip-reading) [39] leads to alternative neural strategies for the processing of this information. Moreover, none of the prior studies have included early-deaf people who are not signers but used rather spoken language and explore the distinct effect of linguistic experience and auditory deprivation on visual crossmodal plasticity. The goal of the present study was to compare neural responses to both emblems and pantomimes between early-deaf and hearing individuals, and for the first time to relate these to behavioral performance. Given the lack of convergence in previous studies, we expected that combined behavioral and fMRI results

might seize compensatory brain plasticity in early-deaf individuals, independently of their primary mean of communication. To test our hypothesis, we measured the fMRI bold response to emblems and pantomimes recognition in early-deaf individuals who used or not sign language in comparison to hearing peers.

## Methods

### Participants

Thirty-five French-speaking adults participated in the present study. All the participants provided written informed consent prior to testing and all experiments were performed in accordance with relevant guidelines and regulations. This study was approved by the ethics committee and scientific boards of the Centre de Recherche Interdisciplinaire en Réadaptation du Montréal métropolitain (CRIR) and the Quebec Bio-Imaging Network (QBIN). One deaf and two hearing participants were excluded from the study due to technical problems during fMRI data acquisition. A total of 32 participants were therefore included in the study: 16 early severe-to-profound deaf subjects (11 women, *Mean age ± SD* = 30.25 ± 4.69 years) were compared to 16 hearing participants (12 women, *Mean age ± SD* = 30.31 ± 5.42 years) matched on age, sex, and number of years of education. All subjects had a normal or corrected-to-normal vision and no history of neurological pathology. According to the Edinburgh handedness inventory index [40], five deaf and three hearing participants were left-handed. All participants were administered the Matrix Reasoning subtest of the Weschler Abbreviated Scale of Intelligence (WASI-II) [41], which is a brief evaluation of non-verbal intelligence, namely of nonverbal fluid reasoning [42]. The results showed that both groups performed in the average to the superior level of ability, as indicated by T scores (deaf participants: $M ± SD$ = 57.44 ± 4.85; hearing participants: $M ± SD$ = 62.46 ± 4.45).

Deaf participants had a severe-to-profound hearing loss greater than 77 dB HL ($M ± SD$ = 94.11 ± 9.93) in both ears as determined by certified audiologists. Specifically, 13 participants had a hearing loss greater than 90 dB HL at 500, 1000, 2000, 4000, and 8000 Hz in both ears while two participants were able to detect 500 Hz pure tones presented at 80 dB HL and 77 dB HL in their better ear. Four participants reported having hereditary congenital deafness whereas, for twelve participants, congenital or early deafness was due to unknown etiologies. Eight of the sixteen deaf participants were proficient signers and four of them were native deaf signers in the *Langue des Signes* Québécoise (LSQ). Eight participants had been using hearing aids since childhood, used spoken language only for expression and relied on lip-reading for reception (see Table 1 for detailed information about the participants).

### Stimuli and experimental protocol

The stimuli consisted of 126 point-light animated videos representing 42 emblems (e.g. "*calm down*"), 42 pantomimes (e.g. "*playing guitar*"), and 42 scrambled versions of these biological motions (Fig 1). We carefully controlled point-light stimuli, which allowed us to isolate the effects of biological motion from possible confounding effects such as face and body perception. Point-light also allows us to isolate biological motion processing from more general visual motion perception, by including a control condition in which the starting positions of the points are randomized, but their motion vectors remain the same [19]. Previous studies that used videos and often compared gesture conditions to non-motion control conditions, were thus limited in the interpretation of their results [43].

An event-related fMRI protocol was split in two runs both presented in random order across participants. The stimulation task was implemented on Psychopy with Python 3.4. Each run of six-minutes comprised 63 different videos (21 stimuli of each category). Safety

**Table 1. Demographic and clinical data for the 16 deaf participants.**

| Subject | Sex | Etiology | Age | Hearing aid | Hearing loss: left ear/right ear (dBHL) | Primary language | WASI T-score | Handedness |
|---|---|---|---|---|---|---|---|---|
| 1 | M | Unknown | 36 | No | 100/100 | Sign | 54 | R |
| 2 | F | Genetic | 22 | No | 105/110 | Sign (native) | 63 | R |
| 3 | F | Genetic | 25 | No | 90/90 | Sign (native) | 52 | L |
| 4 | M | Unknown | 36 | No | 90/90 | Sign | 58 | L |
| 5 | F | Unknown | 29 | Yes | 115/110 | Spoken | 68 | R |
| 6 | F | Genetic | 25 | Yes | 93/95 | Spoken | 58 | L |
| 7 | F | Unknown | 29 | No | 90/90 | Sign (native) | 62 | R |
| 8 | M | Unknown | 25 | Yes | 87/92 | Spoken | 50 | R |
| 9 | F | Unknown | 33 | Yes | 103/102 | Spoken | 58 | L |
| 10 | F | Unknown | 34 | Yes | 106/106 | Spoken | 60 | R |
| 11 | F | Genetic | 36 | No | 90/90 | Sign | 52 | R |
| 12 | F | Unknown | 28 | Yes | 93/92 | Spoken | 62 | R |
| 13 | F | Unknown | 37 | Yes | 78/77 | Spoken | 58 | R |
| 14 | F | Unknown | 28 | No | 97/95 | Sign (native) | 58 | R |
| 15 | M | Unknown | 31 | No | 90/90 | Sign | 60 | L |
| 16 | M | Unknown | 30 | Yes | 101/106 | Spoken | 49 | R |

instructions and imaging sequences were explained to the participants to familiarize them with the fMRI environment. The participants all performed a training trial of the biological motion task before the fMRI session. The instructions were presented before each run and the participants had to press a button once they were done reading them. Each video lasted between two to four seconds and was followed by an inter-stimulus interval randomly varying from two to ten seconds. Biological motion stimuli were projected on a screen at the back of the scanner and were presented to the participants through a mirror attached to the MRI head coil–at approximately 12 cm away from the eyes. With an fMRI-safe button response pad, participants were asked to press as fast and as accurately as possible with the correct button (1: whether the video was a human motion with no communicative content (pantomimes condition), 2: a human motion with communicative content (emblems condition) or 3: a non-human motion (scrambled condition)). Participants performed the task with their dominant hand. Accuracy (percentage of correct answers) and response time were measured.

## Statistical analysis on behavioral data

Accuracy and response time measures of the biological motion task were analyzed using a 3 x 2 repeated-measures ANOVA with point-light conditions (emblems, pantomimes and scrambled) as within-subjects factor and group (deaf and hearing) as a between-subject factor. A Greenhouse-Geisser correction was applied to the degrees of freedom and to the significance level to prevent the disrespect of the sphericity assumption. Because the duration of the videos varied and ranged from two to four seconds in each point-light condition, a two-way ANOVA was conducted to examine the influence of run (1 and 2) and point-light conditions (emblems, pantomimes, and scrambled stimuli) on stimuli duration. On average, duration time of the videos was 3047.62 ms ($SD$ = 740.013) for emblems, 2857.14 ms ($SD$ = 792.82) for pantomines, and 2380.95 ms ($SD$ = 734.28) for scrambled stimuli. The main effect of the run was not significant ($F(2, 120)$ = .000; $p$ = 1.00), suggesting that the two runs were similar in stimulus duration. However, there was a significant main effect of point-light conditions ($F(2, 120)$ = 10.43; $p$ < .001; $\eta^2$ = .148) suggesting that the duration of the stimuli differed among to the conditions. The interaction was not significant ($F(2, 120)$ = .000; $p$ > .05). Bonferroni *post hoc* tests

showed that stimulus duration was significantly higher for emblems than for pantomimes ($p < .001$) and scrambled stimuli ($p < .001$) whereas no significant differences were found between pantomimes and scrambled stimuli ($p > .05$). Consequently, these results show that emblem stimuli were significantly longer than the other two conditions. To address this, participants' response time was transformed into a global mean response time for all point-light conditions across groups. Each response time was then weighted by the duration of the video and multiplied by the global mean.

## fMRI acquisition parameters

Whole-brain anatomical and functional images were acquired using a 3-T Trio Tim system (Siemens Magnetom, Erlangen, Germany) equipped with a 32-channel head coil. Multislice T2*-weighted fMRI images were obtained with a gradient echo-planar sequence using axial slice orientation (TR = 2200 ms, TE = 30 ms, FA = 90˚, 35 transverse slices, 3.2 mm slice thickness, FoV = 192 x 192 mm$^2$, matrix size = 64 x 64 x 35, voxel size = 3 x 3 x 3.2 mm$^3$). Head movements were restrained using foam pads. A structural T1-weighted MPRAGE sequence was also acquired for all participants (voxel size = 1×1×1 mm$^3$, matrix size = 240 x 256, TR = 2.300 ms, TE = 2.98 ms, TI = 900 ms, FoV 256, 192 slices).

## Processing of functional images

The fMRI data were analyzed using SPM 12 in a Matlab environment (Statistical Parametric Mapping, Centre for Neuroimaging, London, UK, http://www.fil.ion.ucl.ac.uk/spm, Matlab 8.5 (Mathworks, Natick, MA, USA). Standard preprocessing was performed (realignment, co-registration of functional and anatomical data). At the step of normalization, two distinct anatomical templates were created using DARTEL [44] (Diffeomorphic Anatomical Registration Through Exponentiated Linear algebra), namely, a template designed for hearing participants and another designed for deaf participants. Both templates were created separately for each group and they have been respectively normalized to the MNI template. A groupwise registration using DARTEL was chosen to reduce possible deformations of the structures that are more difficult to match to the average template based on neurotypical individuals [44]. The DARTEL templates are especially relevant given that previous studies have shown significant structural alterations between deaf and hearing individuals [45]. Finally, spatial smoothing was performed (8-mm FWHM) after which linear contrast images were calculated to test main effects in each participant for each condition ([Emblems], [Pantomimes], [Scrambled]). These linear contrasts generated statistical parametric maps [SPM(T)].

## Statistical analyses of fMRI images

The General Linear Model used for the first level analysis (fixed effects) predicted whole brain bold response at each voxel as the dependent variable and conditions: emblem, pantomime and scrambled point-light movements as predictor factors of change in bold response. The resulting individual contrasts, testing the significance of model estimated *betas* for each condition were smoothed and entered for the second level analysis.

For the second level analysis (random effects), we used the GLM with a full factorial design to estimate the effect of point-light stimulation conditions between groups. The within-subject factor is condition: Emblem, Pantomime, Scrambled. The between-subject factor is group: Deaf vs Hearing. Model estimates resulted in contrasts for the main effect of conditions, the main effect of group and the interactions between groups and conditions. We also tested difference contrasts to assess specific directions of change using *t-contrasts*. As for biological motion, it was calculated as scrambled contrast subtracted from the sum

(Emblems + Pantomimes) in each group separately. We contrasted biological motion, emblems and pantomimes between groups to assess specific activations in the deaf group for the processing of both types of gestures. To assess the relation between change in brain activity and behavioral measures, we used a full factorial design group (2) by condition (3) and difference in response times as a covariate in the model.

**Within-group differences.** *t*-contrasts were calculated for the difference between conditions ([Emblems > Pantomimes], [Emblems > Scrambled], [Pantomimes > Emblems], [Pantomimes > Scrambled], [Scrambled > Emblems], [Scrambled > Pantomimes]) using the false discovery rate FDR-correction for multiple comparisons at a probability level $p < 0.05$. Significantly activated areas are presented as threshold maps in the results section. A conjunction contrast (conjunction null hypothesis) characterized brain areas jointly activated by the contrasts [Emblems + Pantomimes] in both groups.

**Between-group analyses.** To examine group effect on bold activity change for each condition separately, we calculated *t*-contrasts for group differences by condition ([Deaf > NH] x [Emblems], [Deaf > NH] x [Pantomimes], ([NH > Deaf] x [Emblems], [NH > Deaf] x [Pantomimes]) using FDR-correction for multiple comparisons, at $p < 0.05$. The contrast for comparison of brain activations during biological motion processing [(Emblems + Pantomimes)-scrambled] between deaf and NH participants allowed very strict control of low-level stimulus features [19,46].

Finally, we used a general linear model to predict bold change in brain activity with group (deaf vs hearing) and condition (Emblem, Pantomime and Scrambled) as between and within-subjects' factors respectively. The model included the behavioral differences ([Emblems—Pantomimes]) for response times measure as a covariate. The resulting *F-contrast* accounted for significant covariance between groups/condition and behavioral differences using FDR-correction for multiple comparisons, at $p < 0.05$.

## Results

### Behavioral data

Deaf and hearing groups were equivalent with regards to age ($t(30) = .035$, $p = .682$), number of years of education ($t(30) = 1.965$, $p = .06$), or on their performance on the fluid reasoning subtest ($t(30) = 2.32$, $p = .43$). We performed separate repeated-measures 3 x 2 ANOVAs with both accuracy and response times as the dependent variable. The analysis of correct responses showed a significant main effect of point-light condition ($F(1.93, 57.81) = 95.57$; $p < .001$; $\eta^2 = .76$), no main effect of group ($F(1,30) = .04$; $p = .85$) and no significant interaction ($F(1.93, 57.81) = 3.08$; $p > .05$). On average, deaf participants recognized 73.38% ($SD = 5.33$) of emblems correctly, 81.94% ($SD = 6.59$) of pantomimes and 99.62% ($SD = 0.40$) of scrambled stimuli as compared to respectively 68.94% ($SD = 0.24$), 87.69% (SD = 5.11) and 99.56% (SD = 0.65) for hearing participants. Bonferroni *post hoc* tests demonstrated that all the participants were more accurate in the scrambled condition in comparison to the pantomimes and emblems conditions and more accurate in the pantomimes condition than they were in the emblems condition ($p < .001$ for all differences).

The analysis of response times showed a significant main effects of point-light condition ($F(1.66, 49.88) = 37.69$; $p < .001$; $\eta^2 = .56$), no significant main effect of group ($F(1, 30) = 0.14$; $p = .71$) and, a significant Group × Condition interaction ($F(1.66, 49.88) = 4.63$; $p < .05$; $\eta^2 = .13$) (see Fig 1D). Bonferroni *post hoc* tests demonstrated that the deaf and hearing participants were fastest at identifying the scrambled condition in comparison to the pantomimes and emblems, respectively ($p < .001$ for all differences). Only hearing participants exhibited a

# Biological Motion – Scrambled

**Fig 2. fMRI data.** The conjunction of cortical activations implicated in biological motion processing [(Emblems + Pantomimes)—scrambled] by the group, deaf (Red) and hearing participants (Blue), Overlap (Purple).

significant difference between the pantomimes and the emblems conditions, with faster responses for pantomimes ($p < .001$).

## fMRI data

All results reported as significant in this section survived a threshold of whole-brain $p < .05$ voxel-wise threshold, FWE-corrected. Anatomical labels for active regions are the most probable based on the Harvard-Oxford Cortical Atlas.

**Biological versus scrambled motion.**  We first examined the areas significantly activated by biological motion relative to the scrambled condition [(Emblems + Pantomimes)-scrambled] in each group. As expected, the analyses revealed an overlap in the regions involved in the human action recognition network between the deaf and hearing participants (see Fig 2). Both groups showed extensive bilateral activations that included posterior temporal-occipital regions including V5, pSTS, EBA, and FBA, parietal regions including the right SMG and bilateral SPL; frontal lobe regions including bilateral IFG, frontal operculum/insula, precentral

**Table 2. Brain regions showing significant activations for the conjunction of biological motion (emblems and pantomimes)-scrambled in each group.**

| Anatomical region | Hemi | Cluster size | T | x | y | z | p-FEW corr (p < .05) | Other areas including Distance (mm) |
|---|---|---|---|---|---|---|---|---|
| | | **Deaf** | | | | | | |
| Precentral | L | 4412 | 13.63 | -54 | 2 | 43 | .000 | Postcentral (4.58) |
| | | | | | | | | Frontal mid (10.25) |
| Fusiform | L | 5543 | 12.62 | -39 | -43 | -20 | .000 | Temporal inf (2.45) |
| | | | | | | | | Cerebelum 6 (7.35) |
| Parietal inf | R | 65 | 5.52 | 30 | -46 | 49 | .001 | Parietal inf (1.00) |
| | | | | | | | | Postcentral (4.58) |
| Thalamus | L | 82 | 5.34 | -12 | -16 | 7 | .003 | Pallidum (11.70) |
| | | | | | | | | Caudate (12.04) |
| Thalamus | R | 6 | 4.74 | 6 | -22 | -11 | .026 | Lingual (10.05) |
| | | | | | | | | Parahippocampal (10.82) |
| Fusiform | R | 3 | 4.66 | 36 | -4 | -41 | .034 | Temporal inf (2.24) |
| | | | | | | | | Temporal mid (6.71) |
| | | **Hearing** | | | | | | |
| Fusiform | R | 1766 | 12.12 | 39 | -43 | -20 | .000 | Temporal inf (5.10) |
| | | | | | | | | Cerebelum 6 (5.83) |
| Temporal mid | L | 3874 | 11.80 | -51 | -70 | 1 | .000 | Occipital mid (2.45) |
| | | | | | | | | Occipital inf (5.10) |
| Insula | R | 1356 | 10.97 | 30 | 26 | 1 | .000 | Frontal inf tri (4.58) |
| | | | | | | | | Putamen (5.74) |
| Cerebelum 7b | L | 473 | 7.58 | -12 | -73 | -44 | .000 | Cerebelum 8 (2.24) |
| | | | | | | | | Cerebelum crus2 (5.00) |
| Thalamus | L | 167 | 6.14 | -9 | -16 | 4 | .000 | Thalamus R (11.00) |
| | | | | | | | | Pallidum (13.45) |
| Parietal inf | R | 136 | 6.10 | 27 | -49 | 49 | .000 | Parietal sup (1.73) |
| | | | | | | | | Postcentral (5.20) |

MNI coordinates (x, y, z) of the significant clusters are given, along with the corresponding brain region for this cluster and the other areas included in each cluster, with distance (mm).

gyrus, middle frontal gyrus, and SMA; and the thalamus bilaterally (see Table 2 for locations of peak activations). Extensive cerebellar activity was observed as well.

**Between-group analyses.** Beyond these areas of overlap, some areas showed significant activation only for the deaf group for the biological motion conditions [(Emblems + Pantomimes)-scrambled]. The deaf group showed a significantly stronger bilateral response than the hearing participants in the STG, including the planum temporale (BA 22) and the primary and secondary auditory cortex (BA 41, 42) (see Fig 3B and Table 3). Additionally, only deaf individuals showed activation in the basal ganglia (specifically globus pallidus and the head of the caudate nucleus), and greater extent of activation than hearing individuals in the cerebellum (see Fig 3). In the hearing group, no brain region was found to be more activated than the deaf group (see Table 3).

Brain responses to emblems and pantomimes individually were examined (Deaf > Hearing x [Emblems]; Deaf > Hearing x [Pantomimes]; Hearing > Deaf x [Emblems]; Hearing > Deaf x [Pantomimes]). Again, the deaf group showed a significantly stronger bilateral response to hearing participants in the STG, including the planum temporale (BA 22) and the primary and the secondary auditory cortex (BA 41, 42) (see Fig 4 and Table 3). Notably, the deaf group

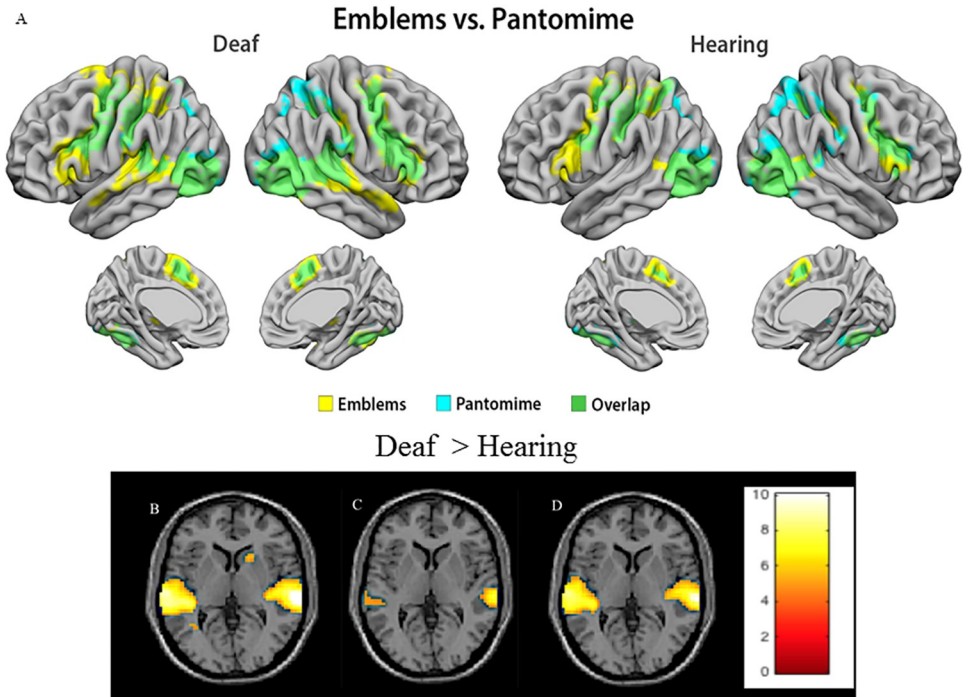

**Fig 3. fMRI data.** (A) The cortical activations implicated in Emblems only (Yellow), Pantomime only (Blue), and the Overlap (Green) by the group. (B) Significant difference between deaf and hearing participants in the biological motion condition, the image in the maximum global coordinate (66.0–28.0 7.0). (C) Significant difference between deaf and hearing participants in the pantomime condition, the image in the maximum global coordinate (66.0–28.0 7.0). (D) Significant difference between deaf and hearing participants in the emblem condition, the image in the maximum peak activation at coordinates (66.0–28.0 7.0). Color scale represents *T* values (B,C, D).

showed a stronger bilateral response for the emblems condition than for the pantomimes condition, including voxels mostly in the planum temporale and in the primary auditory cortex.

**Laterality differences in the deaf group.** We further investigated whether there were laterality differences within the STG clusters activated uniquely in deaf people. Pairwise comparisons were carried out between the average activity (maximum global coordinate, *66.0–28.0 7.0*) in the STG, in both hemispheres in all point-light conditions. The results showed a significant difference in signal strength between the right and the left STG, both in the combined biological motion condition (Emblems + Pantomimes), ($t$ (16) = -8.42, $p$ < .0001 (Right: $M \pm SD$ = 2.31 ± 1.07; Left: $M \pm SD$ = 1.00 ± .91)) as well as in the emblems condition ($t$ (16) = -5.31, $p$ < .0001 (Right: $M \pm SD$ = 2.31 ± .99; Left: $M \pm SD$ = 1.22 ± .85)). A rightward asymmetry was found during processing of scrambled motion and emblems but no difference was found between the hemispheres in the pantomime condition ($t$ (16) = -1.41, $p$ > .15 (Right: $M \pm SD$ = 2.15 ± 1.24; Left: $M \pm SD$ = 1.61 ± .96)). Of interest, an extensive activation of the STG was found in the emblems condition in contrast to the scrambled and pantomimes conditions. The peak activation was located in the primary auditory cortex.

**Covariance with behavioral performance.** As demonstrated earlier, the behavioral results suggest that there was a significant interaction between hearing status with participants' response times (Fig 1). Therefore, the way this behavioral difference [Emblems—Pantomimes] translated into neural activations in the deaf group was explored. A factorial model with group (2 levels) and conditions (3 levels) was used in a whole-brain analysis with behavioral differences (Emblems-Pantomimes response times factored out) as covariates. We found a

**Table 3. Brain regions showing significant activations for the contrast of Deaf > Hearing in each point-light condition.**

*Deaf>Hearing*

| Anatomical region | Hemi | Cluster size | T | x | y | z | *p*-FEW corr (*p* < .05) | Other areas including Distance (mm) |
|---|---|---|---|---|---|---|---|---|
| | | **Biological motion** | | | | | | |
| Temporal sup | R | 969 | 12.32 | 66 | -28 | 7 | .000 | Temporal mid (6.71) |
| | | | | | | | | Supramarginal (11.18) |
| Temporal sup | L | 916 | 10.20 | -54 | -34 | 10 | .000 | Temporal mid (2.00) |
| | | | | | | | | Supramarginal (8.94) |
| Precentral | L | 69 | 6.46 | -57 | -1 | 43 | .001 | Postcentral (1.73) |
| | | | | | | | | Frontal mid (13.96) |
| Caudate | R | 26 | 5.27 | 18 | 17 | 4 | .007 | Putamen (3.00) |
| | | | | | | | | Pallidum (9.00) |
| Occipital Mid | L | 32 | 5.15 | -33 | -58 | 7 | .005 | Calcarine (7.07) |
| | | | | | | | | Precuneus (7.35) |
| Cerebelum 8 | L | 5 | 4.75 | -3 | -61 | -32 | .005 | Vermis 8 (1.41) |
| | | | | | | | | Vermis 9 (3.16) |
| Precentral | R | 5 | 4.75 | 57 | 8 | 37 | .005 | Frontal inf oper (6.16) |
| | | | | | | | | Frontal mid (6.78) |
| | | Emblems | | | | | | |
| Temporal sup | R | 755 | 10.04 | 66 | -25 | 4 | .000 | Temporal Mid (6.08) |
| | | | | | | | | Rolandic oper (11.36) |
| Temporal sup | L | 819 | 8.80 | -60 | -31 | 7 | .000 | Supramarginal (9.90) |
| Precentral | L | 22 | 5.50 | -54 | 2 | 43 | .008 | Postcentral (4.58) |
| | | | | | | | | Frontal Mid (10.25) |
| Caudate | R | 2 | 4.58 | 18 | 17 | 4 | .035 | Putamen (3.00) |
| | | | | | | | | Pallidum (9.00) |
| | | Pantomimes | | | | | | |
| Temporal sup | R | 213 | 7.79 | 66 | -28 | 7 | .001 | Temporal mid (6.71) |
| | | | | | | | | Supramarginal (11.18) |
| Temporal sup | L | 144 | 6.10 | -51 | -37 | 10 | .002 | Temporal mid (2.45) |
| | | | | | | | | Rolandic oper (9.49) |

MNI coordinates (x, y, z) of the significant cluster are given, along with the corresponding brain region for this cluster and the other areas included in each cluster, with distance (mm).

significant covariation between behavioral measures and brain responses in the bilateral STG and the left precentral gyrus (see Fig 4 and Table 4). Brain activity time series in bilateral STG areas were calculated and an independent correlation analysis using Pearson correlation coefficients was carried out to specify the relation between the behavioral measures (response times) and the cerebral activations triggered in the left and right STG. Results indicate that the activation of the STG could predict response times, in the right hemisphere ($r$ = .36, $p$ = .04, $R^2$ = .13) and marginally in the left hemisphere ($r$ = .35, $p$ = .05, $R^2$ = .12). This finding suggests that, for the deaf individuals, stronger activation of the STG during the biological motion task leads to faster response times. Correlation analysis was also conducted on the left precentral gyrus to determine if behavioral results could be predicted by the cortical activity in this region. No significant correlation was found. This was true for the relationship between the peak activity in the precentral gyrus and response times ($r$ = -.37, p > .05) in deaf individuals.

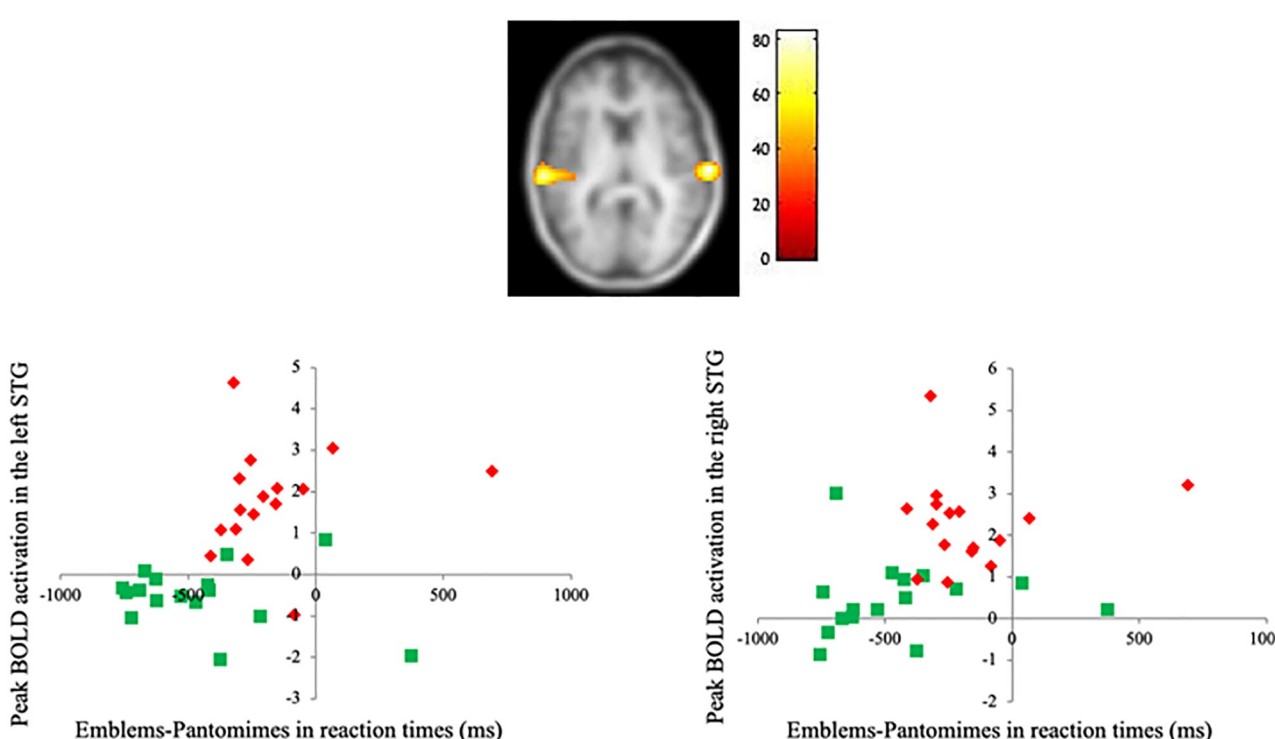

**Fig 4. fMRI data.** Covariation between cortical activity triggered by biological motion (Emblems—Pantomimes) and behavioral discrepancy (on reaction times) in the deaf group only. MNI coordinates for global maximum (66.0–28.0 7.0). Graphs: Correlation plots of the blood oxygen level-dependent Emblems-Pantomimes responses in this region against reaction times (RT). Each data point represents a single subject, Red for the deaf group and Green for the hearing group. Color scale represents *F* values.

**Table 4. Brain regions showing significant activations for the main effect of group with reaction time.**

| Anatomical region | Hemi | Cluster size | F | x | y | z | *p*-FEW corr (*p* < .05) | Other areas including Distance (mm) |
|---|---|---|---|---|---|---|---|---|
| Temporal sup | R | 108 | 82.32 | 66 | -28 | 7 | .000 | Temporal mid (6.71) |
| | | | | | | | | Supramarginal (11.18) |
| Temporal sup | L | 140 | 71.98 | -63 | -31 | 7 | .000 | Supramarginal (9.95) |
| Precentral | L | 14 | 51.40 | -57 | -1 | 46 | .007 | Postcentral (2.45) |
| | | | | | | | | Frontal mid (12.88) |

MNI coordinates (x, y, z) of the significant cluster are given, along with the corresponding brain region for this cluster and the other areas included in each cluster, with distance (mm).

## Discussion

The main goal of the present study was to combine, for the first time, behavioral and neuroimaging measures of emblems and pantomimes gesture recognition, between early-deaf and hearing individuals. In previous studies, inconsistent imaging results were found. A hypoactivation was reported in some cerebral regions involved in the human action network, namely the IPL and the IFG, by two studies investigating the observation of pantomimes in native deaf signers [21,33]. These findings were explained by the predominant use of the visual modality in deaf individuals, not only to support their daily life, but also because of their extensive use of sign language. The latter could be seen as a training in human gestures decoding. The

authors argue that this training could make native deaf signers less sensitive to human gestures and thus result in a cortical hypoactivation [33]. More recently, a study looked at congenitally deaf individuals who were native signers [34]. With a pantomime's judgment task, the authors concluded that there was a robust activation of the human action network in individuals who experienced auditory deprivation in addition to using sign language. However, in this study, no relationship was found between deafs' linguistic experience and the strength of the cortical activations within the human action recognition network [34]. The present study confirms that there is an overlap in deaf and hearing individuals' cortical activation network in response to biological motion processing. Both groups showed similar activations in the expected regions [25], that is, occipital, parietal, temporal, and inferior frontal regions during emblems and pantomimes recognition.

More importantly, the present results provide behavioral and neural evidence in favor of compensatory visual cross-modal activity experienced by early deaf people. As some previous studies [14,34,37], we found significant bilateral activations of the STG, including the primary auditory cortex in the deaf group. Our findings corroborate previous work in the literature. Indeed, there are well-established associations between animal and human data [47] showing that deafness can lead to enhanced visual abilities [6,48], thus implying a cross-modal reorganization process where the visual modality recruits the auditory cortex [4,49,50]. However, the evidence is unclear as to whether deafness can lead to both enhanced behavioral performance and a cross-modal activation of the primary auditory cortex by other sensory modalities or higher cognitive functions [1]. Moreover, the literature on the possible behavioral enhancements experienced by deaf individuals is characterized by results that are both heterogeneous and inconsistent. This can be attributed to a variety of factors, such as sample characteristics [48]. Indeed, variables such as the amount of residual hearing, the onset of deafness or etiology of deafness are known to influence the extent of cerebral plasticity [13,51]. Thus, a majority of studies have specifically examined deaf native signers [51], while these deaf individuals represent only a small percentage of the deaf population [52]. Overall, previous results cannot be generalized, and it is therefore complex to have a clear understanding of deaf individuals' cross-modal reorganization. In our study, differences were found between the behavioral performance and the cortical activation of regions altered by auditory deprivation in deaf compared to hearing participants. The results suggest that early-deaf individuals showed greater sensitivity to the processing of human action than hearing individuals. Specifically, deaf individuals identified emblems as fast as pantomimes in comparison to their hearing peers. These behavioral differences were directly correlated with the bilateral activation of the STG. These results differ from those of previous studies reporting the recruitment of auditory areas in the processing of emblems [37] but not of pantomimes [34], and those reporting no behavioral differences between deaf and hearing participants [34,37]. Additionally, a significant correlation was found between STG activations and response times. This correlation could suggest that the extent of STG recruitment in deaf individuals depends on their capacity to detect emblems more rapidly than pantomimes. This result is consistent with the previous literature showing that enhanced visual performances in deaf individuals are usually related to shorter reaction times rather than to accuracy [5], but must be replicated for exhaustive interpretation.

Furthermore, emblems overall led to more extensive bilateral activations than pantomimes in deaf individuals, especially in the STG (including planum temporale and primary auditory cortex). The activation of the primary auditory cortex, followed by the posterior region of the STG, involved in the dorsal pathway of language processing [53–55], suggests that emblems are more prone to be processed as linguistic material by early-deaf individuals. The linguistic processing of emblems, supported by the activation of the left STG, was reported in a study on prelingual deaf adults who were native signers [37]. According to the authors, the linguistic

processing of emblems is sustained by a leftward hemispheric asymmetry found in deaf signers in comparison to hearing participants. However, several neuroimaging studies propose that language processing implies a collaboration of both left and right pathways, as well as a cortico-sub-cortical network [53]. In addition, the language network in the right hemisphere is classically related to the visual abilities involved in language processing [56] and explains the STG rightward asymmetry during recognition of visually communicative emblems by the deaf group.

The fMRI analyses performed in the present study addressed the implications of auditory deprivation and linguistic experience on visual biological motion processing. All our deaf participants presented profound-to-severe congenital deafness, but while half of them were proficient in sign language (four were native deaf signers), the other half was using spoken language as a first language. While not formally tested, the robustness of the cortical activations in the human action network suggests an absence of any linguistic experience effect. A particularly interesting finding of the present study is that the differences in human action processing are better explained by an effect of auditory deprivation since all the deaf participants experienced a bilateral activation of the STG. In future studies, a larger sample size of deaf individuals would be needed since deafness related factors are known to influence brain plasticity (e.g. deafness duration, amount of residual hearing, prior use of hearing aids) and should be considered in the analyses [13,51].

Functional and behavioral correlates converge to a human action sensitivity following early-deafness deprivation. This sensitivity does not appear to be modulated by linguistic experience but rather by auditory deprivation. Thus, the present findings are of importance not only because they contribute to the understanding of the visual cross-modal plasticity phenomenon in the deaf population, but also because they offer new avenues of research for rehabilitation strategies that would be better adapted to the daily effects of deafness.

## Acknowledgments

The authors declared no competing interests. All data generated or analyzed during this study are included in this published article. We are grateful to the individuals who volunteered for this research and to the staff at the Functional Neuroimaging Unit for testing assistance. We also thank Vanessa Hadid for helpful discussions about data and analysis.

## Author Contributions

**Conceptualization:** Alexandria Muise-Hennessey, Aaron J. Newman.

**Data curation:** Marie Simon, Latifa Lazzouni.

**Formal analysis:** Marie Simon, Latifa Lazzouni.

**Funding acquisition:** Emma Campbell.

**Methodology:** Marie Simon, Latifa Lazzouni, Alexandria Muise-Hennessey, Aaron J. Newman, François Champoux.

**Project administration:** Franco Lepore.

**Resources:** Aaron J. Newman, François Champoux.

**Supervision:** Aaron J. Newman, François Champoux, Franco Lepore.

**Visualization:** Marie Simon.

**Writing – original draft:** Marie Simon, Latifa Lazzouni.

**Writing – review & editing:** Emma Campbell, Audrey Delcenserie, Aaron J. Newman, Fran-çois Champoux, Franco Lepore.

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
