## [Decision Letter · Decision Letter 0]

4 Feb 2020

PONE-D-19-34925

Enhancement of Visual Biological Motion Recognition in Early-Deaf Adults: Functional and Behavioral Correlates

PLOS ONE

Dear Mrs Simon,

Thank you for submitting your manuscript to PLOS ONE. After careful consideration, we feel that it has merit but does not fully meet PLOS ONE’s publication criteria as it currently stands. Therefore, we invite you to submit a revised version of the manuscript that addresses the points raised during the review process.

We would appreciate receiving your revised manuscript by Mar 20 2020 11:59PM. To enhance the reproducibility of your results, we recommend that if applicable you deposit your laboratory protocols in protocols.io, where a protocol can be assigned its own identifier (DOI) such that it can be cited independently in the future. For instructions see: http://journals.plos.org/plosone/s/submission-guidelines#loc-laboratory-protocols

We look forward to receiving your revised manuscript.

Kind regards,

Paul Hinckley Delano, Ph.D.

Academic Editor

PLOS ONE

Journal Requirements:

"This research was supported by the Canada Research Chair Program (F.L.), the Natural Sciences and Engineering Research Council of Canada (F.L.), and a grant from Med-EL Elektromed (A.J.N., F.L., and F.C.) and from Quebec Bio-Imaging Network (M.S). "           

3. Please provide further details regarding how participants were recruited, including the participant recruitment date.

Reviewers' comments:

Reviewer's Responses to Questions

**Comments to the Author**

1. Is the manuscript technically sound, and do the data support the conclusions?

Reviewer #1: Yes

Reviewer #2: Yes

2. Has the statistical analysis been performed appropriately and rigorously? 

Reviewer #1: Yes

Reviewer #2: Yes

3. Have the authors made all data underlying the findings in their manuscript fully available?

Reviewer #1: Yes

Reviewer #2: Yes

4. Is the manuscript presented in an intelligible fashion and written in standard English?

Reviewer #1: Yes

Reviewer #2: Yes

5. Review Comments to the Author

Reviewer #1: The work contrast the recognition of different inferred movements on deaf and hearing population. The study aims to study is rooted not only in exploring cortical processing related to these movements and population characteristics, but also to link this cortical processing with behavioral differences.

In general, the article concludes that minor behavioral differences are found, which are weakly related to STG activity. Cortical activity presents evident differences between groups and conditions.

This is a well written article with a robust experiment design. Methodologically speaking it presents a classic approach properly applied.

However, there are still some room for improvement. The article may improve a few things to facilitate reader’s comprehension, and in my personal opinion, this article may use some different methodological approaches to improve the results they based on in part of the discussion. I think that authors have better results than what they think, but the technique used, does not allow them to catch such results entirely. Finally, there are some methodological decisions that are not argued and they should be. Main reasons, is to improve the understanding of the logic behind those calls, but also to improves readers’ experience and allow people who are naïve to statistics to understand the justifications of the method used.

Specifically:

1. Some results are given in methods sections, which I don’t clearly see why aren’t they in results section.

2. I highly suggest to change “condition” for a meaningful term such a movements or anything that suits authors idea of what they are measuring through those three conditions. This will improve readers’ experience.

3. In results given within methods section, mean +- SD was not reported. Please complete the statistical reports.

4. Authors indicates that they used “global mean response”. I did not understand properly this normalization with the information given in the text. I search information about it, and I find many different conceptualizations for the same concept. As such, I would highly appreciate to include a detailed explanation of what was made in order to increase replicability of this work. Particularly the weighting method is not mentioned at all. Also, no cite for this method, or rationale behind it is given. As it is currently given, it is not easy to understand the properties of this normalization or the reasons to use it.

5. Also, it is not clear when this normalization is used, as in Figure 1 Reaction times are ploted raw.

6. It is not justified why using the differences in reaction time (unclear if normalized or not) as covariates. I also did not understand in which statistical model they were introduce as covariates (which was the dependent variable). Since it is mentioned as start of new paragraph, I don’t understand if it refers to previous or following analyses. I assume, it refers to repeated measures ANOVA; however, this cannot be assessed with ANCOVA, as there is not properly developed repeated measures ANCOVA. This is usually assessed using Mixed Linear Models. In any case, there is no justification to include it in any of them as covariate. I would appreciate a clarification of this point.

7. In line 244 authors say that they will use a correlation (without describing which method specifically) using an F-test. From results I induce that the authors used Pearson correlation, which used r-statistic instead of F statistic. F-statistic is used to evaluate the significance of a linear model, using t-statistic for each coefficient. Please clarify what are the authors exactly applying.

8. In line 261-2 authors cite Figure 1E, but it only has until D.

9. Figure 1 Caption appears 2 times in text body.

10. In Table 2, authors wrote “most significant”. This is vague. Please explain the criteria by which those results were included and why some others were excluded of the report.

11. In line 350, authors wrote “STG significantly predicted response time” base on (what I induce) a correlation. This is not appropriate. Correlations do not allow to talk about predictions or estimations, as coefficients are not estimated. Strictly speaking in this study only estimations can be made as there is no train and new data logic. I don’t mind the misusage of prediction as it may contribute to improves readers’ understanding of the work (most people in biological areas misused the concept), but then coefficients must be reported.

12. In Figure 4 please include a color legend to improve reader’s experience.

13. Authors should consider the usage of F-test in correlations presented in Figure 4, considering that here authors imply a hypothesis that neural activity produce behavior (which makes total sense). In that case, it is better suited a linear model. Also, and considering the plots, particularly for left hemisphere, I would advise authors to fit a linear model and to remove outliers base on bonferroni’s outlier test (is model based, so authors must fit first the model, remove outliers and fit again). Author may also try to fit a model in y=b0+b(Reaction Time)+b(Reaction Time)squared. This would allow to capture the parabolic like pattern that this curve has in left hemisphere in deaf group. It is likely that increase in activity plateaus at some point, which would be consistent with non-linear modeling. This similar pattern seem to occur in NH for right STG. I would also encourage to include other relevant variables in the model which may contribute to reduce the noise. Personally, I think that the method used is not taking the best of the results obtained.

14. Finally, squared Rs derived from correlations are too weak to establish conclusive results. As such, depending on how linear modeling goes, authors may use bootstrap or any cross validation technique to evaluate if fit is low but consistent. Otherwise, I would expect to moderate discussion considering that this result may not be obtained if the study is replicated.

Reviewer #2: The authors study behavioral and fMRI measures of emblems and pantomines gestures recognition, between early-deaf and hearing individuals.

The total number of participants is 32 (16 early-deaf and 16 hearing subjects matched on age, sex and number of education years).

The article is very clear and well written. The methodology is sound, with stimuli that were carefully prepared and the results were appropiately analyzed. Results are well presented and very interesting.

A main finding is the greater sensitivity to the processing of human action in early-deaf invididuals (with a processing of emblems as fast as patomines) that is better explained by an effect of auditory deprivation. These behavioral differences where directly correlated with a bilateral activation of the STG. Also a correlation was found between these activations and response time, suggesting that the extent of STG recruitment in deaf subjects depends on the capacity that have to detect emblems more rapidly than pantomine.

My only concern is the size of the sample. If the paper is accepted, the authors should discuss the possible effects of using such a small sample... for example, there were no differences for the primary language of the deaf individuals (sign, native sign and spoken)... could this be due to the use of too small a sample?

6. PLOS authors have the option to publish the peer review history of their article (what does this mean?). If published, this will include your full peer review and any attached files.

Reviewer #1: No

Reviewer #2: No

---

## [Author Response · Author response to Decision Letter 0]

3 Apr 2020

We are happy to enclose a revised version of the manuscript PONE-D-19-34925 entitled “Enhancement of Visual Biological Motion Recognition in Early-Deaf Adults: Functional and Behavioral Correlates” for consideration in Plos One. 

We have carefully taken all the reviewer's suggestions into consideration to prepare a revised version of the manuscript, which also required modifications in the text and a revised figure. We truly feel the reviewers have contributed to improving the quality and clarity of the manuscript. Therefore, we would like to thank both reviewers for their precise and useful comments and suggestions as well as for their time spent reviewing our work. 

Below, you can find our responses to the comments of Reviewers 1 and 2 as well as actions taken in the the revised manuscript. 

We hope that you will find our revised manuscript satisfactory for publication and would be very proud to be given the opportunity to represent your journal. 

Report of Reviewer #1

The work contrasts the recognition of different inferred movements on deaf and hearing population. The study aims to study is rooted not only in exploring cortical processing related to these movements and population characteristics, but also to link this cortical processing with behavioral differences.

In general, the article concludes that minor behavioral differences are found, which are weakly related to STG activity. Cortical activity presents evident differences between groups and conditions.

This is a well written article with a robust experiment design. Methodologically speaking it presents a classic approach properly applied. 

However, there are still some room for improvement. The article may improve a few things to facilitate reader’s comprehension, and in my personal opinion, this article may use some different methodological approaches to improve the results they based on in part of the discussion. I think that authors have better results than what they think, but the technique used, does not allow them to catch such results entirely. Finally, there are some methodological decisions that are not argued, and they should be. Main reasons are to improve the understanding of the logic behind those calls, but also to improves readers’ experience and allow people who are naïve to statistics to understand the justifications of the method used.

Specifically:

1. Some results are given in methods sections, which I don’t clearly see why aren’t they in results section.

Response: Thank you for your pertinent comment. As you mentioned, some results were given in the method section. Thus, we removed the following sentence: “Deaf and hearing groups were equivalent with regards to age (t(30) = .035, p = .682), number of years of education (t(30) = 1.965, p = .06), or on their performance on the fluid reasoning subtest (t(30) = 2.32, p = .43)”, previously L.161 to L255, in the beginning of the method.

2. I highly suggest to change “condition” for a meaningful term such a movements or anything that suits authors idea of what they are measuring through those three conditions. This will improve readers’ experience.

Response: We thank the reviewer for the suggestion and have adjusted the term “condition” by “point-light condition” in all the manuscript. The new terminology has been inspired by, previous studies such as. Campbell et al., 2011. 

3. In results given within methods section, mean +- SD was not reported. Please complete the statistical reports.

Response: Thank you for your observation. We added all the mean+/- SD missing in the first draft of the manuscript. 

4. Authors indicates that they used “global mean response”. I did not understand properly this normalization with the information given in the text. I search information about it, and I find many different conceptualizations for the same concept. As such, I would highly appreciate to include a detailed explanation of what was made in order to increase replicability of this work. Particularly the weighting method is not mentioned at all. Also, no cite for this method, or rationale behind it is given. As it is currently given, it is not easy to understand the properties of this normalization or the reasons to use it.

Response: As explained in the manuscript (starting L.198), we used normalized reaction times because 1) we found a significant difference between the time duration of the three types of stimuli, 2) despite the instruction to press as fast and accurately as possible, all the participant (deaf and hearing) waiting for the end of the stimulus to answer. Therefore, we normalized reaction times based on a calculation below: each individual's response time to a video has been divided by the duration of that specific video then multiplied by the global mean, which represents the mean duration of all 126 videos combined (L212).

5. Also, it is not clear when this normalization is used, as in Figure 1 Reaction times are plotted raw.

Response: As explained above, we chose to normalize reaction times because the duration of the videos varied from two to four seconds in each point-light condition. Thus, Figure 1 represents normalized reaction times and all following statistical tests on the reaction time measure have been performed with these. We are confident that they represent an optimal measure of behavior during the recognition of biological motion.

6. It is not justified why using the differences in reaction time (unclear if normalized or not) as covariates. I also did not understand in which statistical model they were introduce as covariates (which was the dependent variable). Since it is mentioned as start of new paragraph, I don’t understand if it refers to previous or following analyses. I assume, it refers to repeated measures ANOVA; however, this cannot be assessed with ANCOVA, as there is not properly developed repeated measures ANCOVA. This is usually assessed using Mixed Linear Models. In any case, there is no justification to include it in any of them as covariate. I would appreciate a clarification of this point.

Response: The difference in reaction times for emblems and pantomimes is not normalized because reaction times were already normalized to total mean response time for the three conditions and video duration. There is also a significant difference in response times for both groups between conditions (behavioral differences statistical analysis). This difference accounts for the additional semantic processing time of emblems compared to pantomimes. The difference in RTs is then introduced in the 2nd level (fMRI statistical analysis in SPM) regression model as a covariate to test correlation between reaction time change and brain signal change in Emblem-pantomime contrast across subjects. 

7. In line 244 authors say that they will use a correlation (without describing which method specifically) using an F-test. From results I induce that the authors used Pearson correlation, which used r-statistic instead of F statistic. F-statistic is used to evaluate the significance of a linear model, using t-statistic for each coefficient. Please clarify what are the authors exactly applying.

Response: In this case, the 2nd level analysis in SPM was applied using a multiple regression model introducing signal change for emblem-pantomime contrast as the dependent variable and the difference in reaction time between emblem and pantomime as a covariate for both groups, to test for covariation (either positive or negative) between behavioral performance and brain signal change.

The t statistic is used to test for Emblem-Pantomime contrast and the covariance with RT difference. As such for voxels which pass correction the signal change and performance are significantly correlated. The results in Figure 4 graph represent the relationship between behavioral change and brain activity at the voxel of maximum activity which value was extracted from SPM.

8. In line 261-2 authors cite Figure 1E, but it only has until D.

Response: We thank the reviewer for the suggestion and have corrected this error and replaced “Figure 1E” by “Figure 1D”. 

9. Figure 1 Caption appears 2 times in text body.

Response: Thank you for your comment. We remove one on the legend under the name of “Figure 1” in the main text. 

10. In Table 2, authors wrote “most significant”. This is vague. Please explain the criteria by which those results were included and why some others were excluded of the report.

Response: Thank you for your comment. Indeed, we re-phrased this sentence, instead of “MNI coordinates (x, y, z) of the most significant cluster are given (…)”, 

We suggest: “MNI coordinates (x, y, z) of the significant cluster are given (…).” Thus, we applied this correction to all the MRI legends. 

11. In line 350, authors wrote “STG significantly predicted response time” base on (what I induce) a correlation. This is not appropriate. Correlations do not allow to talk about predictions or estimations, as coefficients are not estimated. Strictly speaking in this study only estimations can be made as there is no train and new data logic. I don’t mind the misusage of prediction as it may contribute to improves readers’ understanding of the work (most people in biological areas misused the concept), but then coefficients must be reported.

Response: We thank the reviewer for the suggestion. We have strictly pursued statistical guidelines for correlations based on Cohen, J. 1988. Statistical Power Analysis for the Behavioral Sciences, 2nd Edition. Routledge. Thus, L. 372, we have reported all classical values: p, r and R2 as coefficient. We re-phrased this sentence, instead of “STG significantly predicted response time”. 

We suggest: “STG could predict response time”. 

12. In Figure 4 please include a color legend to improve reader’s experience.

Response: Thank you, we agree with your suggestion. We have proceeded to minor changes in the Fig.4 to include a color legend. We believe this has helped to clarify the figure.

13. Authors should consider the usage of F-test in correlations presented in Figure 4, considering that here authors imply a hypothesis that neural activity produce behavior (which makes total sense). In that case, it is better suited a linear model. Also, and considering the plots, particularly for left hemisphere, I would advise authors to fit a linear model and to remove outliers base on bonferroni’s outlier test (is model based, so authors must fit first the model, remove outliers and fit again). Author may also try to fit a model in y=b0+b (Reaction Time)+b(Reaction Time)squared. This would allow to capture the parabolic like pattern that this curve has in left hemisphere in deaf group. It is likely that increase in activity plateaus at some point, which would be consistent with non-linear modeling. This similar pattern seems to occur in NH for right STG. I would also encourage to include other relevant variables in the model which may contribute to reduce the noise. Personally, I think that the method used is not taking the best of the results obtained.

Response: As explained above the statistical analysis for covariance between RT difference and brain activity was done in SPM as part of the imaging data statistical analysis. The use of a linear model is interesting but as long as there is a collinearity between brain activation and behavioral performance other variables are introduced instead of STG brain activation. The communication means (sign, aural) seems to predict well the change in RT. With signers showing reduced change in RTs between Emblems and pantomimes. See results:

14. Finally, squared Rs derived from correlations are too weak to establish conclusive results. As such, depending on how linear modeling goes, authors may use bootstrap or any cross-validation technique to evaluate if fit is low but consistent. Otherwise, I would expect to moderate discussion considering that this result may not be obtained if the study is replicated.

Response: We will consider discussing this point in the paper, see L. 440-447. 

Report of Reviewer #2

The authors study behavioral and fMRI measures of emblems and pantomimes gestures recognition, between early-deaf and hearing individuals. The total number of participants is 32 (16 early-deaf and 16 hearing subjects matched on age, sex and number of education years).

The article is very clear and well written. The methodology is sound, with stimuli that were carefully prepared, and the results were appropriately analyzed. Results are well presented and very interesting.

A main finding is the greater sensitivity to the processing of human action in early-deaf individuals (with a processing of emblems as fast as pantomimes) that is better explained by an effect of auditory deprivation. These behavioral differences where directly correlated with a bilateral activation of the STG. Also, a correlation was found between these activations and response time, suggesting that the extent of STG recruitment in deaf subjects depends on the capacity that have to detect emblems more rapidly than pantomime.

My only concern is the size of the sample. If the paper is accepted, the authors should discuss the possible effects of using such a small sample... for example, there were no differences for the primary language of the deaf individuals (sign, native sign and spoken)... could this be due to the use of too small a sample?

Response: Thank you for your comment, we agree that this information should be mentioned as one of the conclusions and have modified the manuscript to account for your comment. To express this, we suggest adjusting the wording with the following: “In future studies, a larger sample size of deaf individuals would be needed since deafness related factors influencing brain plasticity (e.g. deafness duration, amount of residual hearing, prior use of hearing aids) and should be considered in the analyses (13,54).” However, we are very confident with our result showing that behavioral difference was correlated to a bilateral activation in the STG in all our early deaf participants, independently of the primary mean of communication. 

Finally, we want to report to the reviewer 2 the challenges that are linked to the recruitment of deaf individuals, especially for imaging research who are, at some point, disconnected to their daily-life preoccupation. Therefore, the specific characteristics of this population make it difficult to access through the usual recruitment “channels” despite support, in our case, from a major deaf rehabilitation center in Montreal. In previous studies, the number of deaf adults tested in imaging varied between 6 to 53 (see for example, Meyer et al., 2007 or Shibata 2007, Lepore et al., 2010 or Smithenaar et al., 2016). We recently performed a power analysis to estimate the optimal sample. The analysis suggests that a sample of 25 deaf participants is required to perform linear regression linked to deafness heterogeneity, thus, we hope that futures studies will replicate our main results while enriching them with regression analyzes.

---

## [Decision Letter · Decision Letter 1]

1 May 2020

PONE-D-19-34925R1

Enhancement of Visual Biological Motion Recognition in Early-Deaf Adults: Functional and Behavioral Correlates

PLOS ONE

Dear Mrs Simon,

Thank you for submitting your manuscript to PLOS ONE. After careful consideration, we feel that it has merit but does not fully meet PLOS ONE’s publication criteria as it currently stands. Therefore, we invite you to submit a revised version of the manuscript that addresses the points raised during the review process.

Please respond to all concerns raised by reviewer 1.

We would appreciate receiving your revised manuscript by Jun 15 2020 11:59PM. To enhance the reproducibility of your results, we recommend that if applicable you deposit your laboratory protocols in protocols.io, where a protocol can be assigned its own identifier (DOI) such that it can be cited independently in the future. For instructions see: http://journals.plos.org/plosone/s/submission-guidelines#loc-laboratory-protocols

We look forward to receiving your revised manuscript.

Kind regards,

Paul Hinckley Delano, Ph.D.

Academic Editor

PLOS ONE

Reviewers' comments:

Reviewer's Responses to Questions

**Comments to the Author**

1. If the authors have adequately addressed your comments raised in a previous round of review and you feel that this manuscript is now acceptable for publication, you may indicate that here to bypass the “Comments to the Author” section, enter your conflict of interest statement in the “Confidential to Editor” section, and submit your "Accept" recommendation.

Reviewer #1: All comments have been addressed

Reviewer #2: All comments have been addressed

2. Is the manuscript technically sound, and do the data support the conclusions?

Reviewer #1: Yes

Reviewer #2: Yes

3. Has the statistical analysis been performed appropriately and rigorously? 

Reviewer #1: Yes

Reviewer #2: Yes

4. Have the authors made all data underlying the findings in their manuscript fully available?

Reviewer #1: Yes

Reviewer #2: Yes

5. Is the manuscript presented in an intelligible fashion and written in standard English?

Reviewer #1: Yes

Reviewer #2: Yes

6. Review Comments to the Author

Reviewer #1: Authors have addressed most of the comments given. However, there are still one major confusion which limits strongly the replicability of the reported results, and two minor comments about the Figures.

1- In line 251 to 254, the GLM method is briefly explained. First, there is no mention to the specific GLM used. From comments of the authors, I must assume that a Generalized Mixed Linear Model was used. Second, authors mention correlations (without mention which kind) analysis as synonym of GLM, which is not the case. Third, there is no mention to the structure of the GLMM used. Which variables were of first level, second level, were interaction specified? which ones, and so on. This is relevant, as introducing random effects and cross-level interactions are likely to produce overfit. Fourth, authors report Pearson r instead of beta coefficient, while saying that they used GLM. GLM uses F-statistic, not r-statistic. One can obtain r.-statistic from GLM; however, is not a proper results report and omit the most relevant information, like the beta value.

Since authors have been so consistent in using as synonym correlations, GLM and GLMM, I reviewed in detail the SPM documentation and documentation used for tutorial and divulgation purposes. In many of these documents it is stated that GLM implemented in SPM can be used to asses t-test, correlations, linear modeling, and generalized linear models. Correlations and t-test are not GLM; however, one can used particular cases of GLM to answer almost the same questions answered by t-test and correlations. In the case of ANOVA and ANCOVA, they are fairly the same test. However, in the case of correlation and t-test they are not equivalent to GLM. Discrepancies will respond mostly to assumption violation and sample size used. Suggesting them as synonym only contributes to obscure the methods used and limiting replicability.

I understand the source of the confusion. However, current statistical report limits strongly replicability and metanalytic work. To solve the current confusion, please:

a-Refer always to GLM always instead of correlation.

b-In method sections specify properly the structure of the GLM model used. Which variables in 1st and 2nd level, if any interaction was introduced, and which variables were introduced as fixed or random effects.

c-Please check the SPM output if beta values (also referred as coefficients) are reported. It may be the case that if you are using random effects, these will not be reported, or they would not be of easy access. In that case, please remove the r-statistics of the current results report (p-values are not estimated based on such r-statistic, therefor is misleading to report it).

d-If possible, report the F-statistic from which GLM model significance is estimated (current draft does not report GLM significance, I recommend including it if possible). Also, if possible, include the t statistic usually used for GLMM coefficients.

2- In Figure 3 B, C, D and in Figure 4, there is a yellow to white color scale without units. There is no explanation either of what these scales represents. Please include the vriable to which is referring and the units of such variables.

3- Figure 1 D, standard unit for milliseconds is ms, not MSEC. Capital M states Mega, which means *10^6, while m means *10^-3. Sec is an informal abbreviation; s is the standard unit for seconds.

Reviewer #2: (No Response)

7. PLOS authors have the option to publish the peer review history of their article (what does this mean?). If published, this will include your full peer review and any attached files.

Reviewer #1: No

Reviewer #2: No

---

## [Author Response · Author response to Decision Letter 1]

14 Jun 2020

We are happy to enclose a second revised version of the manuscript PONE-D-19-34925 entitled “Enhancement of Visual Biological Motion Recognition in Early-Deaf Adults: Functional and Behavioral Correlates” for consideration in Plos One. 

We have carefully taken all the reviewer 1 suggestions into consideration to prepare a revised version of the manuscript, which also required modifications in the text and a revised figure. We truly feel the reviewer have contributed to improving the quality and clarity of the manuscript. 

Below, you can find our responses to the comments of Reviewer 1 as well as actions taken in the revised manuscript. 

We hope that you will find our revised manuscript satisfactory for publication and would be very proud to be given the opportunity to represent your journal. 

Report of Reviewer #1

Reviewer #1: Authors have addressed most of the comments given. However, there are still one major confusion which limits strongly the replicability of the reported results, and two minor comments about the Figures.

1- In line 251 to 254, the GLM method is briefly explained. First, there is no mention to the specific GLM used. From comments of the authors, I must assume that a Generalized Mixed Linear Model was used. Second, authors mention correlations (without mention which kind) analysis as synonym of GLM, which is not the case. Third, there is no mention to the structure of the GLMM used. Which variables were of first level, second level, were interaction specified? which ones, and so on. This is relevant, as introducing random effects and cross-level interactions are likely to produce overfit. Fourth, authors report Pearson r instead of beta coefficient, while saying that they used GLM. GLM uses F-statistic, not r-statistic. One can obtain r.-statistic from GLM; however, is not a proper results report and omit the most relevant information, like the beta value.

Since authors have been so consistent in using as synonym correlations, GLM and GLMM, I reviewed in detail the SPM documentation and documentation used for tutorial and divulgation purposes. In many of these documents it is stated that GLM implemented in SPM can be used to asses t-test, correlations, linear modeling, and generalized linear models. Correlations and t-test are not GLM; however, one can used particular cases of GLM to answer almost the same questions answered by t-test and correlations. In the case of ANOVA and ANCOVA, they are fairly the same test. However, in the case of correlation and t-test they are not equivalent to GLM. Discrepancies will respond mostly to assumption violation and sample size used. Suggesting them as synonym only contributes to obscure the methods used and limiting replicability.

I understand the source of the confusion. However, current statistical report limits strongly replicability and metanalytic work. To solve the current confusion, please:

a-Refer always to GLM always instead of correlation.

Response: Thank you for your pertinent comment, this is now done when it applies. 

b-In method sections specify properly the structure of the GLM model used. Which variables in 1st and 2nd level, if any interaction was introduced, and which variables were introduced as fixed or random effects.

Response: Thank you, we agree with your suggestion. We have proceeded to minor changes This section is now part of the manuscript: 

The General Linear Model for first level analysis predicted the bold response as the dependent variable and conditions: emblem, pantomime and scrambled point light movements as predictor factors of change in bold response. 

The resulting individual contrasts (comparing estimated Betas to 0 using one tailed t-test) for each condition were smoothed and entered for the second level analysis. 

For the second level analysis we used a full factorial design to estimate the effect of point light stimulation conditions between groups. The within subject factor is condition: Emblem, Pantomime, Scrambled. The between subject factor is group: Deaf vs controls.

Model estimates resulted in contrasts for the main effect of conditions, the main effect of group and the interactions between groups and conditions.

We also tested difference contrasts to assess specific directions of change using t-tests. Biological motion contrast was calculated as scrambled subtracted from the sum (Emblems + Pantomimes) in each group separately.

We contrasted biological motion, emblems and pantomimes between groups to assess specific activations in the deaf group for the processing of both types of gestures.

To assess the relation between change in brain activity and behavioral measures, we used a full factorial design group (2) by condition (3) and difference in response times as a covariate in the model.

c-Please check the SPM output if beta values (also referred as coefficients) are reported. It may be the case that if you are using random effects, these will not be reported, or they would not be of easy access. In that case, please remove the r-statistics of the current results report (p-values are not estimated based on such r-statistic, therefor is misleading to report it).

Response: Once the model of covariation between changes in brain activity between groups and conditions and behavioral measures was estimated, areas showing significant covaration with performance measures were identified in bilateral STG regions.

Resulting brain activity time series in bilateral STG areas were calculated (at peak activity) and an independent correlation analysis was carried out using Pearson correlation coefficients to specify the relation between the behavioral measures (response times) and the cerebral activations in the left and right STG. Correlation results reported in the manuscript are related to this analysis. 

This is now precised in the manuscript by adding a section.

d-If possible, report the F-statistic from which GLM model significance is estimated (current draft does not report GLM significance, I recommend including it if possible). Also, if possible, include the t statistic usually used for GLMM coefficients.

Response:

F statistic for the main effect of group: F=22.5960 (p=0.05 FDR corrected) 

F statistic for the main effect of condition: F=13.6698 (p=0.05 FDR corrected)

F statistic for the covariance with behavioral performance F=35.3696 (p=0.05 FDR corrected)

Minor

2- In Figure 3 B, C, D and in Figure 4, there is a yellow to white color scale without units. There is no explanation either of what these scales represent. Please include the variable to which is referring and the units of such variables.

Response: Thank you for your observation, we added the missing information. 

Figure 3: color scale represents T values

Figure 4: color scale represents F values

3- Figure 1 D, standard unit for milliseconds is ms, not MSEC. Capital M states Mega, which means *10^6, while m means *10^-3. Sec is an informal abbreviation; s is the standard unit for seconds.

Response: We thank the reviewer for the suggestion and have corrected this error in the Figure 1D.

---

## [Editor Report · Decision Letter 2]

16 Jun 2020

PONE-D-19-34925R2

Enhancement of Visual Biological Motion Recognition in Early-Deaf Adults: Functional and Behavioral Correlates

PLOS ONE

Dear Dr. Simon,

Thank you for submitting your manuscript to PLOS ONE. After careful consideration, we feel that it has merit but does not fully meet PLOS ONE’s publication criteria as it currently stands. Therefore, we invite you to submit a revised version of the manuscript that addresses the points raised during the review process.

As suggested by one reviewer, give a rigorous description of all statistical tests and fix minor issues in Figures.

We look forward to receiving your revised manuscript.

Kind regards,

Paul Hinckley Delano, Ph.D.

Academic Editor

PLOS ONE

---

## [Author Response · Author response to Decision Letter 2]

13 Jul 2020

We have fixed minor issues in Figures

---

## [Editor Report · Decision Letter 3]

15 Jul 2020

Enhancement of Visual Biological Motion Recognition in Early-Deaf Adults: Functional and Behavioral Correlates

PONE-D-19-34925R3

Dear Dr. Simon,

We’re pleased to inform you that your manuscript has been judged scientifically suitable for publication and will be formally accepted for publication once it meets all outstanding technical requirements.

Kind regards,

Paul Hinckley Delano, Ph.D.

Academic Editor

PLOS ONE
---

## [Editor Report · Acceptance letter]

20 Jul 2020

PONE-D-19-34925R3 

Enhancement of Visual Biological Motion Recognition in Early-Deaf Adults: Functional and Behavioral Correlates 

Dear Dr. Simon:

I'm pleased to inform you that your manuscript has been deemed suitable for publication in PLOS ONE. Congratulations! Your manuscript is now with our production department. 

Kind regards, 

on behalf of

Dr. Paul Hinckley Delano 

Academic Editor

PLOS ONE